# High Extra Virgin Olive Oil Consumption Is Linked to a Lower Prevalence of NAFLD with a Prominent Effect in Obese Subjects: Results from the MICOL Study

**DOI:** 10.3390/nu15214673

**Published:** 2023-11-04

**Authors:** Calogero Claudio. Tedesco, Caterina Bonfiglio, Maria Notarnicola, Maria Rendina, Antonino Castellaneta, Alfredo Di Leo, Gianluigi Giannelli, Luigi Fontana

**Affiliations:** 1Data Science Unit, National Institute of Gastroenterology, IRCCS “Saverio de Bellis” Research Hospital, 70013 Castellana Grotte, Italy; 2Laboratory of Epidemiology and Statistics, National Institute of Gastroenterology, IRCCS “Saverio de Bellis” Research Hospital, 70013 Castellana Grotte, Italy; catia.bonfiglio@irccsdebellis.it; 3Laboratory of Nutritional Biochemistry, National Institute of Gastroenterology, IRCCS “Saverio de Bellis” Research Hospital, 70013 Castellana Grotte, Italy; maria.notarnicola@irccsdebellis.it; 4Gastroenterology and Digestive Endoscopy, University Hospital, Policlinico of Bari, 70100 Bari, Italy; mariarendina@virgilio.it; 5Gastroenterology Unit, Department of Precision and Regenerative Medicine and Ionian Area (DiMePRe-J), University of Bari Aldo Moro, 70100 Bari, Italy; antocastellaneta@hotmail.com (A.C.); alfredo.dileo@uniba.it (A.D.L.); 6Scientific Direction, National Institute of Gastroenterology, IRCCS “Saverio de Bellis” Research Hospital, 70013 Castellana Grotte, Italy; gianluigi.giannelli@irccsdebellis.it; 7Charles Perkins Centre, Faculty of Medicine and Health, The University of Sydney, Sydney, NSW 2006, Australia; luigi.fontana@sydney.edu.au; 8Department of Endocrinology, Royal Prince Alfred Hospital, Sydney, NSW 2006, Australia

**Keywords:** olive oil, Mediterranean diet, fatty liver, obesity, odds ratio, relative risk reduction

## Abstract

Extra virgin olive oil (EVOO) has healthy benefits for noncommunicable diseases (NCDs). However, limited evidence is available about the effects of liver disease and non-alcoholic fatty liver disease (NAFLD). We evaluate whether dose-increased consumption of EVOO is associated with a lower prevalence of NAFLD and if these effects vary based on body weight. The study included 2436 subjects with a 33% NAFLD prevalence. Daily EVOO was categorized into tertiles: low (0–24 g/day), moderate (25–37 g/day), and high consumption (>37 g/day). Subjects were also classified by body mass index (BMI) as normo-weight (18.5–24.9), overweight (25–29.9), and obese (≥30). Logistic regression analysis was applied to calculate odds ratios (ORs) for NAFLD, considering a 20-gram increment in EVOO intake and accounting for EVOO categories combined with BMI classes. The ORs were 0.83 (0.74;0.93) C.I. *p* = 0.0018 for continuous EVOO, 0.89 (0.69;1.15) C.I. *p* = 0.37, and 0.73 (0.55;0.97) C.I. *p* = 0.03 for moderate and high consumption, respectively, when compared to low consumption. Overall, the percent relative risk reductions (RRR) for NAFLD from low to high EVOO consumption were 18% (16.4%;19.2%) C.I. and 26% (25%;27.4%) C.I. in overweight and obese subjects. High EVOO consumption is associated with a reduced risk of NAFLD. This effect is amplified in overweight subjects and even more in obese subjects.

## 1. Introduction

The incidence of non-alcoholic fatty liver disease (NAFLD) has been rising globally, linked to increasing obesity rates [1]. Excessive caloric intake and sedentary lifestyles are major contributors to NAFLD, which, if not properly treated, can evolve into a more severe condition, such as non-alcoholic steatohepatitis (NASH), characterized by necroinflammation and faster fibrosis [2]. The only therapeutic approaches are lifestyle modification and weight loss, which consistently reduce liver steatosis and fibrosis [3,4,5]. However, emerging evidence underscores the importance of dietary composition related to NAFLD. The Mediterranean diet (MD) is considered the diet of choice for treating NAFLD as recommended by EASL–EASD–EASO guidelines [6,7,8]. Moreover, several studies have demonstrated that certain components of the diet can either promote or inhibit the progression of NAFLD. Foods rich in refined carbohydrates, fructose, saturated fatty acids, and trans fatty acids have been associated with liver steatosis and inflammation, exacerbating NAFLD [9,10]. Conversely, monounsaturated fatty acids, omega-3 polyunsaturated fatty acids, and dietary fibers have been found to inhibit liver fat accumulation by reducing the process of de novo hepatic lipogenesis [11,12,13]. The role of monounsaturated fat intake in NAFLD is still controversial, with conflicting results in different studies. Some studies suggest a beneficial effect of foods rich in monounsaturated fats, like olive oil, on reducing liver fat and inflammation [14,15,16]. In contrast, others report a potentially detrimental effect that might be due to an impairment of hepatic mitochondrial fatty acid oxidation that is influenced by energy metabolism and body composition [17,18]. The purpose of this study was to assess whether an increase in extra virgin olive oil consumption is associated with a lower prevalence of NAFLD with a possible differential effect by body weight, independently from other food intake, and multiple sociodemographic and metabolic risk factors in a prospective cohort living in a Mediterranean country.

## 2. Materials and Methods

### 2.1. The MICOL Study

The MICOL Study is a population-based prospective cohort originally recruited to investigate cholelithiasis epidemiology. Subjects were randomly drawn from the electoral list of Castellana Grotte in 1985 and followed up until 2016, with four repeated assessments about every 8 years. In 2005–2006, a young random sample of subjects (PANEL Study) aged 30–50 years was added to balance the cohort aging. Extensive data on sociodemographic factors, anthropometric measurements, health status, and lifestyle were collected through questionnaires, with medical history confirmed through verbal interviews. Additionally, participants completed a Food Frequency Questionnaire, underwent liver ultrasound examinations, and provided biological samples. The baseline was set at the third cohort assessment (2005–2006), encompassing a heterogeneous population with a wide age range. A total of 2436 subjects, with 52% men aged between 30 and 89 years, were included in the analysis, with an NAFLD prevalence of 33%. All participants provided signed informed consent. Full details of the study have been previously published [19,20].

### 2.2. Outcome, Clinical, and Dietary Data

Liver steatosis was assessed using ultrasound imaging (Hitachi H21 Vision, Hitachi Medical Corporation, Tokyo, Japan) with a 3.5 MHz transducer. The presence or absence of hyperechogenic liver parenchyma was used to determine the presence of steatosis [6,21]. NAFLD was defined as the presence of steatosis from unknown causes. Therefore, subjects with secondary causes of steatosis were excluded from the study, including those with fatty liver disease due to excessive ethanol consumption (AFLD definition > 30 g/day for men and >20 g/day for women), the use of steatogenic drugs, viral hepatitis B based on an ELISA serum test for surface antigen (HBsAg), and viral hepatitis C based on an antibody (anti-HCV) search confirmed by the Strip Immunoblot Assay RIBA HCV 2.0. Covariates were selected from known risk factors. Sociodemographic information, medical history, family history of chronic diseases, smoking status, and dietary habits were obtained from questionnaires administered during the baseline assessment and follow-up visits. Anthropometric measurements were taken using standardized procedures. Weight was measured using the SECA^®^ body composition analyzer (Seca Deutschland, Hamburg Germany), to the nearest 0.1 kg, and height was measured using a wall-mounted stadiometer. Waist and hip circumference were measured with the patient’s feet joined, abdominal muscles relaxed, and arms hanging down the body. Body mass index (BMI) was calculated as weight divided by height squared (kg/m^2^). Blood samples were collected after at least 12 h of fasting, and routine biochemical assays were performed using standard laboratory methods. Measurements included total bilirubin, glucose, gamma-glutamyl transferase (GGT), aspartate aminotransferase (AST), alanine aminotransferase (ALT), high-density lipoprotein cholesterol (HDL-C), low-density lipoprotein cholesterol (LDL-C), and triglycerides. Participants’ dietary intake was assessed using the European Prospective Investigation into Cancer and Nutrition (EPIC) food frequency questionnaire (FFQ). This questionnaire asked participants to estimate the frequency and quantity of specific foods consumed, reported in times per week, month, or year. The FFQ also included information on the average quantity of each food consumed per day. To aid in estimating portion sizes, participants were provided with photos showing examples of different portion sizes (small, medium, and large) for various foods. Total energy intake was calculated by summing the kilocalories from each food item. Food items were grouped into eight MD components, similarly to Trichopoulou et al. [22], without alcohol intake. These components, expressed in grams per day, included extra virgin olive oil, legumes, cereals (including bread and potatoes), fruits, vegetables, fish, meat and meat products, and milk and dairy products. All procedures were conducted in accordance with the ethical standards of the institutional research committee (IRCCS Saverio de Bellis Research) and followed the principles outlined in the 1964 Helsinki Declaration. The MICOL study received ethical committee approval (DDG-CE-347/1984, DDG-CE-453/1991, DDG-CE-589/2004, and DDG-CE-782/2013) for the ethical conduct of the research.

### 2.3. Statistical Analysis

Population characteristics were summarized according to the type of variable. Continuous normal data were reported as mean ± standard deviation (SD) or median with an interquartile range for skewed data. Categorical variables were reported with frequencies and percentages. Student’s *t*-test, Wilcoxon rank sum test, or Chi-square test were used to evaluate differences between two independent groups if the variable has a normal, skewed, or categorical distribution, respectively. Food groups were analyzed as continuous variables or categorized into tertiles as low consumption, moderate consumption, and high consumption. Analyses were also stratified according to BMI classes to evaluate different effects of food by BMI level and compute BMI* food interaction. A total of 16 subjects were found to be underweight (<18.5); thus, due to this small subgroup, it was not possible to estimate any effect, so they were excluded from the analysis. In total, three BMI classes were chosen: normo-weight (18.5–24.9), overweight (25–29.9), and obese (>30). The association of food consumption with the odds of having NAFLD was evaluated by logistic regression analysis, adjusting for covariates. The first model (model 1) included adjustments for anthropometric and lifestyle characteristics: age, gender, BMI, educational level, living together, and smoking habit. The second model (model 2) included full adjustment with the risk factors and blood measures: diabetes, systolic blood pressure (SBP), diastolic blood pressure (DBP), high-density lipoprotein cholesterol (HDL-C), low-density lipoprotein cholesterol (LDL-C), triglycerides, bilirubin, glucose, gamma-glutamyl transferase (GGT), aspartate aminotransferase (AST), and alanine aminotransferase (ALT). Food components were standardized to compare the odds ratio among them. The relative risk reduction (RRR) for NAFLD from low to high food consumption was calculated as 1− (OR_low_/OR_high_) percent if protective food, or vice versa as (OR_high_/OR_low_) − 1 percent if the food increased the risk. Sensitivity analysis was also performed by deriving the subject’s basal metabolic rate (BMR) based on Schofield equations [23] to check under- and over-reporting of food intake.

A *p*-value < 0.05 was considered to be statistically significant. All analyses were performed using SAS software version 9.4 (SAS Institute).

## 3. Results

The descriptive characteristics of the population, according to the disease group, are reported in Table 1. Participants with NAFLD were typically older, predominantly men, with a higher BMI, waist circumference, and blood pressure. Additionally, this group had higher plasma concentrations of triglyceride, glucose, LDL-cholesterol, AST, ALT, and GGT and lower levels of HDL-cholesterol. As regards comorbidity, the NAFLD group also showed a significantly higher rate of hypertension in diabetic subjects.

### 3.1. Role of Mediterranean Food Items as NAFLD Risk Factors

BMI was an important confounder for the association between food items and NAFLD. Higher BMI is not only associated with a greater risk of NAFLD but also with increased consumption of certain food groups. When accounting for BMI alone, some significant associations between foods and NAFLD emerged. Specifically, EVOO showed a significant inverse association with NAFLD OR = 0.78, 95% C.I. (0.71;0.87), *p* < 0.0001, as did vegetables OR = 0.9, 95% C.I. (0.820;0.997), *p* = 0.04, and legumes OR = 0.9, 95% C.I. (0.820;0.993), *p* = 0.036. The dose-response relationship between food consumption and NAFLD odds is reported in Table 2. To determine the independent effects of each food on the risk of developing NAFLD, all food components were simultaneously included in the model, adjusting for BMI and total kilocalorie intake. Olive oil consumption was found to be inversely associated with NAFLD, with an OR = 0.867 and a 95% C.I. (0.761;0.988), *p* = 0.032. On the other hand, refined cereals appeared to be significantly associated with NAFLD in this adjusted model (*p* = 0.02). These findings highlight the independent protective effect of olive oil consumption on NAFLD risk while emphasizing the potential contribution of refined cereals to the development of the disease. The associations of continuous EVOO intake and categorical EVOO intake with NAFLD, adjusting for other notable risk factors, are presented in Table 3. Continuous EVOO intake showed a strong association with NAFLD in both Model 1 OR = 0.79, 95% C.I. (0.71;0.88), and Model 2 OR = 0.83, 95% C.I. (0.74;0.93). In categorical analysis, EVOO intake was divided into tertiles: low consumption (0–24 g/day), moderate consumption (25–37 g/day), and high consumption (>37 g/day), which roughly corresponds to approximately (0–2), (>2–3), and (>3) tablespoons of olive oil per day, respectively. The highest EVOO consumption category showed a 26.5% reduction in the risk of NAFLD (*p* = 0.032) compared to the lowest category after adjusting for covariates in the full Model 2.

Figure 1 illustrates the trend of odds ratios by each specific g/day cutoff of olive oil intake, with a 5-gram step increase, aiming to display the threshold with the maximum benefit. It shows a clear and strong reduction in risk with an intake of EVOO up to 85 g. However, beyond this threshold, the odds ratios begin to rise in a non-significant manner.

### 3.2. Differential Effect of EVOO on Body Weight

Figure 2 presents the stratified log-odds estimates for NAFLD based on EVOO categories and BMI classes, taking into account the adjustments in Model 2. In overweight subjects, there was a reduction in NAFLD risk, with odds ratios decreasing from low EVOO consumption OR = 5.16 to high consumption OR = 4.24, resulting in a relative risk reduction RRR = 18%, 95% C.I. (16.4%;19.2%), *p* = 0.046. Among obese subjects, the difference is even more pronounced, with odds ratios decreasing from low consumption OR = 16.8 to high consumption OR = 12.4 and a RRR = 26%, 95% C.I. (25%;27.4%), *p* = 0.015. In the normal weight group, high versus low consumption showed an opposite trend but was not statistically significant. The overall *p*-interaction was <0.0001, demonstrating a stronger beneficial effect of EVOO consumption in overweight and obese participants. Appendix A also shows the metabolic profile combined with EVOO categories and BMI classes. A higher intake of EVOO is associated with a significant increase in the plasma concentration of HDL-cholesterol in obese and overweight individuals. However, this association is not observed in normo-weight participants. Conversely, ALT concentrations show a significant decrease with higher consumption of EVOO in obese participants but not in normo-weight participants, where AST, GGT, and triglyceride concentrations are significantly increased.

### 3.3. Sensitivity Analysis

Sensitivity analysis was conducted to exclude misreporting subjects based on their basal metabolic rate (BMR), calculated from the Schofield equation reported in Appendix A. Of the initial samples, 141 subjects (5.8%) were identified as under-reporters, while 100 subjects (4.1%) were over-reporters. This resulted in a final sample of 2195 correct reporters for the analysis. The odds ratio for continuous EVOO intake, Model 2 adjusted, remained similar to the previous analysis (Table 3) and statistically significant. The new odds ratio was 0.845, 95% C.I. (0.748;0.954), *p* = 0.006. When combining EVOO categories with BMI classes (3 × 3 combination), the stratified effects of EVOO were modified and remained statistically significant with an overall *p*-interaction < 0.0001, similar to the previous analysis in Figure 2. In the overweight category, the odds ratios were modified to OR = 5.09 for low consumption and OR = 4.62 for high consumption, with a relative risk reduction RRR = 9.1%, 95% C.I. (8.0%;10.2%), *p* = 0.13. In the obese category, the odds ratios were modified to OR = 18.7 for low consumption and OR = 12.9 for high consumption, with a RRR = 31.2%, 95% C.I. (29.5%;32.8%), *p* = 0.02. These results indicate an even greater reduction in NAFLD risk compared to the previous RRR estimate for obese participants. These findings confirm the robustness of the associations between EVOO consumption, BMI, and NAFLD risk, highlighting the significant protective effects of EVOO, particularly among overweight and obese individuals.

## 4. Discussion

The results of this prospective study, involving 2436 middle-aged and older adults, suggest that a higher intake of EVOO is linked to a reduced prevalence of NAFLD and an improved metabolic profile, specifically in individuals who are overweight or obese. However, these benefits were not observed in individuals with a normal weight, where EVOO intake was associated with significant increases in levels of liver transaminases and triglycerides. Our findings also show that the beneficial effect of olive oil is not mediated by other factors such as age, sex, diet, smoking, and various metabolic and sociodemographic variables. Based on these findings, it is recommended that individuals with a BMI higher than 25 include olive oil in their diet as part of the management of NAFLD. A high-calorie diet rich in saturated fats, refined carbohydrates, fructose, and red meat increases the risk of NAFLD [9,24]. Consistently, in our study, we found that a high intake of refined cereals was associated with an increased risk of NAFLD. Conversely, an energy-restricted Mediterranean-like diet consisting of extra virgin olive oil, legumes, vegetables, nuts, fish and whole grains helps prevent NAFLD [6,7,25,26,27,28]. However, not all studies have reported consistent findings [29,30]. Our data suggest that a high intake of EVOO prevents fatty liver, and interestingly, the risk of the disease seems to reach a maximum decrease with an 85 g/day intake of EVOO, equivalent to seven tablespoons/day. This aligns with the findings of the PREDIMED trial, which showed that supplementing the Mediterranean diet with at least four tablespoons of olive oil per day reduced hepatic steatosis from 33% to 8.8% in overweight and obese older adults at high cardiovascular risk [27]. Another randomized trial found that consuming 20 g of olive oil per day led to a lower fatty liver grade in overweight/obese NAFLD patients compared to those consuming the same amount of sunflower oil, independent of cardiometabolic risk factors [31]. Monounsaturated fatty acids, the primary component of EVOO, have been observed to have both protective and detrimental effects on liver steatosis and inflammation in animal studies and cell cultures [18,32]. The impact of these fatty acids on liver health appears to be complex and context-dependent. Our study suggests that this inconsistency may be influenced, in part, by body composition and related metabolic adaptations. Interestingly, in a prospective cohort study involving 3882 elderly Caucasians, increased consumption of animal proteins was initially associated with a higher risk of NAFLD. However, when adjusting for BMI, the statistical significance was lost [33]. Additionally, the rich concentration of polyphenols found in EVOO may play a role in its potential benefits against steatosis. These phytochemicals possess properties that can help reduce oxidative stress, inflammation, and promote liver health, thereby contributing to the potential antisteatotic effects of EVOO [32,34,35,36]. Our study has some limitations. Firstly, the cross-sectional design of the analysis prevents us from drawing definitive conclusions, as in randomized controlled trials with controlled effects. Secondly, there is a possibility of reverse causality, as patients at high risk may have already modified their diet. However, when adjusting for lifestyle characteristics and risk factors, the inverse association between EVOO and NAFLD remained significant. It is important to note that dietary assessment in epidemiological studies can be affected by recall bias and misreporting, which can attenuate the associations. We conducted a sensitivity analysis, which yielded similar results to the initial findings, and in the specific obese category, the association was even stronger.

## 5. Conclusions

In conclusion, our study highlights the potential benefits of consuming olive oil in the context of NAFLD and emphasizes the importance of considering individual weight status in dietary interventions. The impact of olive oil on NAFLD appears to be primarily influenced by adiposity, independent of other factors. These findings have important implications for public health, considering the increasing prevalence of NAFLD. Additional randomized trials are required to understand the mechanisms and establish optimal EVOO intake for NAFLD prevention and management, accounting for individual weight status.

## Figures and Tables

**Figure 1 nutrients-15-04673-f001:**
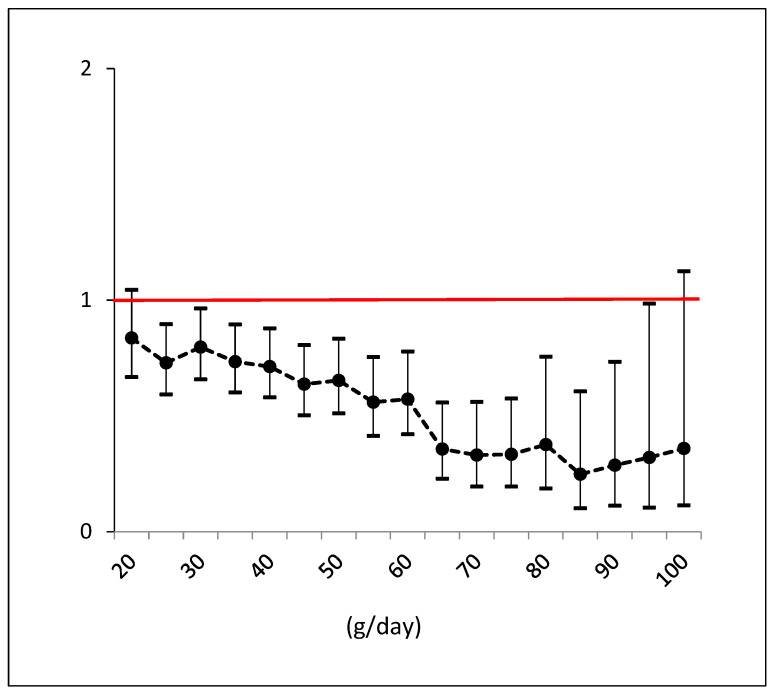
Odds ratios and confidence intervals for NAFLD by each cutoff consumption of EVOO (e.g., >55 g versus <55 g). Black dots are the odds ratios with line intervals. Confidence intervals which do not cross red line are significant effects.

**Figure 2 nutrients-15-04673-f002:**
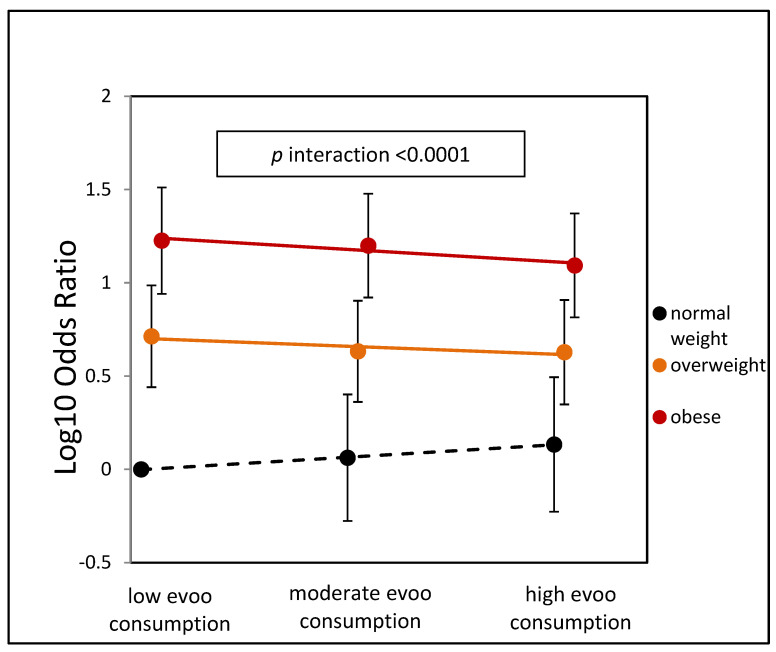
Adjusted odds ratio for NAFLD by olive oil categories and BMI categories. The reference category is low EVOO-normal weight.

**Table 1 nutrients-15-04673-t001:** Summary of the study population.

	No NAFLD	NAFLD	
	*n* = 1628	*n* = 808	*p*-Value
Demographics			
Age (years)	53.8 ± 16	55.6 ± 14	0.004
Male *n* (%)	804 (49)	468 (58)	<0.0001
Educational levels			
Illiterate	59 (3.6)	30 (3.7)	
Primary School	469 (28.9)	254 (31.4)	
Secondary School	495 (30.5)	247 (30.6)	0.6
High School	401 (24.7)	191 (23.6)	
Graduated	200 (12.3)	86 (10.6)	
Living together yes *n* (%)	1215 (74)	647 (80)	0.003
Work classes			
Jobless or Pensioneers	550 (33.8)	298 (36.9)	
Elementary Occupations	584 (35.9)	293 (36.3)	0.47
Craft, Agriculture, and Sales	421 (25.9)	185 (22.9)	
Managers and Professionals	73 (4.5)	32 (4)	
Smoking habit yes *n* (%)	274 (17)	122 (15)	0.27
Anthropometrics			
BMI classes			
Normo-weight	587 (36.1)	47 (5.8)	
Overweight	722 (44.4)	287 (35.5)	<0.0001
Obese	319 (19.6)	474 (58.7)	
Waist (cm)	87 ± 12	100 ± 11.5	<0.0001
Hip (cm)	100 ± 9.2	109 ± 11	<0.0001
Food data			
Extra virgin olive oil (g/day)	31.4 (20.1; 41.8)	31.4 (20.7; 41.8)	0.90
Fruits (g/day)	517.8 (301.3; 928.2)	499.6 (286.4; 895.7)	0.25
Vegetables (g/day)	202.4 (125.8; 310)	196.3 (118.7; 306.2)	0.23
Legumes (g/day)	30.2 (17.1; 45.6)	27.6 (17.1; 45.4)	0.34
Cereals (g/day)	239.2 (164.7; 363.6)	230.8 (151.6; 350.2)	0.08
Fresh Fish (g/day)	30.7 (17.1; 47.2)	29.7 (16.6; 49)	0.64
Total Meat (g/day)	66.1 (43.1; 99)	65.6 (43.4; 100.8)	0.67
Dairy Products (g/day)	201 (110.3; 287.5)	188.7 (104; 279.5)	0.17
Biochemistry			
ALT (U/L)	12 (10; 16)	17 (13; 24)	<0.0001
AST (U/L)	10 (9; 12)	12 (10; 14)	<0.0001
GGT (U/L)	11 (8; 15)	14 (11; 19)	<0.0001
Bilirubin (mg/dL)	0.82 (0.67; 0.99)	0.79 (0.64; 1)	0.08
Glucose (mg/dL)	100 (94; 108)	107 (99; 118)	<0.0001
HDL-C (mg/dL)	52 (44; 63)	44 (39; 53)	<0.0001
LDL-C (mg/dL)	119 (99; 141)	124 (101; 146)	0.008
Triglycerides (mg/dL)	88 (65; 125)	137 (93; 188)	<0.0001
Comorbidities			
Hypertension yes *n* (%)	470 (29)	334 (41)	<0.0001
SBP (mm Hg)	120.8 ± 20	126.8 ± 18.6	<0.0001
DBP (mm Hg)	73.2 ± 9.8	76.8 ± 9.9	<0.0001
Diabetes yes *n* (%)	112 (7)	116 (14)	<0.0001

BMI, body mass index; ALT, alanine aminotransferase; AST, aspartate aminotransferase; GGT, gamma-glutamyl transferase; HDL-C, high-density lipoprotein cholesterol; LDL-C, high-density lipoprotein cholesterol; SBP, systolic blood pressure; DBP, diastolic blood pressure. Reported statistics are: mean ± SD, median (q1; q3), frequency (percentage).

**Table 2 nutrients-15-04673-t002:** Multivariable logistic regression model for NAFLD with all Mediterranean diet components.

	Units	OR *	C.I.		*p*-Value
BMI (kg/m^2^)	(1 kg/m^2^)	1.257	1.228	1.287	<0.0001
EVOO (g/day)	1 sd ≈ 20 g	0.867	0.761	0.988	0.032
Fruits (g/day)	1 sd ≈ 570 g	1.114	0.959	1.294	0.16
Vegetables (g/day)	1 sd ≈ 190 g	0.961	0.857	1.078	0.49
Legumes (g/day)	1 sd ≈ 25 g	0.928	0.831	1.035	0.18
Cereals (g/day)	1 sd ≈ 150 g	1.325	1.044	1.681	0.02
Fresh Fish (g/day)	1 sd ≈ 30 g	1.035	0.927	1.155	0.54
Total Meat (g/day)	1 sd ≈ 54 g	1.062	0.937	1.204	0.34
Milk and Dairy (g/day)	1 sd ≈ 150 g	1.063	0.945	1.197	0.30
Total kilocalories	(unit 500 kcal)	0.810	0.655	1.002	0.052

* ORs were computed per increase by one standard deviation.

**Table 3 nutrients-15-04673-t003:** Multivariable logistic regression model for NAFLD.

	Model 1 *				Model 2 **			
	OR	C.I.		*p*-Value	OR	C.I.		*p*-Value
continuous EVOO ^§^	0.792	0.710	0.880	<0.0001	0.833	0.743	0.934	0.0018
low consumption (ref.)	1	-	-	-	1	-	-	-
moderate consumption	0.835	0.660	1.060	0.139	0.891	0.690	1.151	0.37
high consumption	0.658	0.510	0.860	0.002	0.735	0.555	0.974	0.032

* The NAFLD logistic model adjusted for age, gender, BMI, education level, living together, and smoking. ** NAFLD logistic model adjusted for variables in model 1, including diabetes, sbp, dbp, hdl, ldl, triglycerides, glucose, bilirubin, AST, ALT, and GGT. ^§^ ORs were computed per increase by one standard deviation.

## Data Availability

The data described in the manuscript will not be available for basic research but, upon reasonable request, will be available to the corresponding author and be suitable for new collaborations such as data integration or data pooling.

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
