# Peer review of "High Extra Virgin Olive Oil Consumption Is Linked to a Lower Prevalence of NAFLD with a Prominent Effect in Obese Subjects: Results from the MICOL Study"

_nutrients, 2023, doi:10.3390/nu15214673_

Round 1
Reviewer 1 Report
Comments and Suggestions for Authors
This paper examines the potential benefits of consumption of extra virgin olive oil(EVOO) for NAFLD in a prospective cohort living in a Mediterranean country, it also emphasizes the importance of considering individual weight status in dietary intervention. However, there are several issues that need to be addressed:
1. Table 2 shows the dose-response relationship between the intake of each food and the incidence of NAFLD, and the investigator should specify how often and how much the subject consumed the particular type of food. In addition, studies should indicate that participants' diets should be adjusted according to their BMI at different stages to better account for BMI was an important confounder for foods-NAFLD associations, and further evidence that an energy-restricted Mediterranean food items can help prevent NAFLD.
2. It is mentioned in the article that EVOO intake is inversely associated with the risk of NAFLD. There was a dose-response relationship between EVOO intake and NAFLD. Each 5g increase in intake further reduces the risk of NAFLD until the risk of NAFLD is substantially reduced at levels up to 85g of EVOO. However, the nutritional value of the oil is determined by the amount of unsaturated fatty acids, the more unsaturated fatty acids, the higher the nutritional value. In other words, the amount of EVOO should generally be half or less of normal cooking oil. So whether the daily intake of up to 85g of EVOO has practical significance.
3. Triglyceride is one of the necessary adipokines in the human body. It is normally stored in the liver. Various high-density lipoprotein and low-density lipoprotein are used to adjust adipokines in the human body for fat metabolism. The hallmark of NAFLD is accumulation of triglyceride in the liver. In examining the effects of EVOO in subjects of different weight, it was found that the concentration of triglycerides increased significantly with the increase of EVOO intake in participants of normal weight. However, in overweight and obese participants, triglyceride levels did not decrease as EVOO intake increased. Therefore, whether the article wrote that EVOO intake can reduce the prevalence of NAFLD in overweight or obese people is accurate.
Comments on the Quality of English LanguageMinor editing of English language required
Reviewer 2 Report
Comments and Suggestions for Authors
The search for an optimal nutritional model that would effectively reduce the severity of non-alcoholic fatty liver disease is purposeful and justified. The importance of olive oil in the diet as a factor influencing the functioning of the liver is an important issue and has application significance. However, the study did not address whether the study participants consumed types of olive oil other than extra virgin as sources of, among others, oleic acid. Similarly, estimating the level of adherence to the Mediterranean diet using validated tools (e.g. The Mediterranean-diet score) would help create a regression model taking into account diet as a factor determining liver steatosis. The authors present the impact of extra virgin olive oil consumption on lipid metabolism parameters and indicators of liver function, but no such relationships are presented for parameters directly related to hepatic steatosis, such as the hepatic fat percentage or fatty liver grade.
It should also be explained why an inversely proportional relationship was observed between vegetable consumption and the occurrence of NAFLD despite the lack of significant differences in the consumption of this group of products in people without and with non-alcoholic fatty liver disease.
Additionally, the description of the research methodology should include a method for determining resting energy expenditure. Similarly, the characteristics of people participating in the study should include information about their health, e.g. the presence of type 2 diabetes or hypertension.
Providing information on how many people with normal body weight had GGT and TG blood concentrations above the reference values could explain their increase with increasing EVOO consumption? Additionally, how many normal weight people had NAFLD?
Additional comments:
Figure 1 shows that the risk of NAFLD decreases significantly when consuming more than 60g/d of extra virgin olive oil, not after consuming a much larger amount > 85g/d
In the title of Table 2 "... with all mediterranean components" it is worth adding "... with all mediterranean diet components".
Large literary abbreviations should be included in the description of the parameters (line 143-144).
